# Peer review of "Information Fusion in Autonomous Vehicle Using Artificial Neural Group Key Synchronization"

_sensors, 2022, doi:10.3390/s22041652_

Round 1

Reviewer 1 Report

The comments of this manuscript are presented as follows:
1.In the equation (1)~(4), there are four elements of the overall key sharing delays. Why are the delays the same? If there are just four hypotheses, please explain it in the manuscript.
2.Many of the same references have been cited several times in the paper, for example, the reference [7], the author can replace it with a specific method. In addition, the references cited in the paper are not cited in order, please revise them.
3.The three primary types of security features in Section 4 are not reflected in Section 5. Please add some experiments.
4.In the Table 3, the specific methods represented by references [39], [40], [12], [41] and [42] should be presented.

Author Response

The authors express their gratitude for the constructive comments and suggestions made by the esteemed reviewers. The followings are comment wise responses by the authors.

Reviewer #1:

  1. In the equation (1)~(4), there are four elements of the overall key sharing delays. Why are the delays the same? If there are just four hypotheses, please explain it in the manuscript.

Reply: As per the reviewer’s advice, the equation (1)-(4) are modified in the revised manuscript. Instead of “=” in these equations “+” operator is used. All four delays are explained in the revised manuscript and now it is clearly shown that the delays are not same.

The following paragraphs are incorporated in the revised manuscript to explain these four hypotheses.

When the cars reach an unexpected area, as mentioned in section 3, this approach leverages current key sharing techniques as backup options. In this part, this method will use the area-based key sharing technique to predict the average delay of key sharing, with the current key sharing technique as a backup.  Timevehi, TimeRSU, Timekey center, and  TimeNetork are the four elements of the overall key sharing delay for the one-time key update, as shown in figure 12.

Timevehi  refers to the time it takes for the vehicle to generate the request message, sign it, get the response from the key distribution server, and verify the response. TimeRSU  is the time it takes for the RSU ends to validate the key update request, forward it, get the response message from the key distribution server, and sign it. The delay on the key distribution server is therefore included in Timekey center, which is simplified as verify and response.

TimeNetork  delay on a heterogeneous network comprises delay on the DSRC network between cars and , delay on the Internet between and key distribution servers, and random delay congestion on the link between and key distribution.

  1. Many of the same references have been cited several times in the paper, for example, the reference [7], the author can replace it with a specific method. In addition, the references cited in the paper are not cited in order, please revise them.

Reply: As per the reviewer’s advice, the problem of reference citation is resolved. Also, references are cited by maintaining a proper order.

  1. The three primary types of security features in Section 4 are not reflected in Section 5. Please add some experiments.

Reply: As per the reviewer’s advice, to ensure the security features following experiments are performed and discussed in the results section.

Table 4 shows the outcomes for different τ1 τ2 τ3 N λ  values. Two columns define the minimum and maximum syncing times. The capacity of an adversary to formulate and implement the actions of approved TLTPMs is evaluated by invader syncing.

The arrangements (8-8-8-16-8) and (8-8-8-8-128) give the optimum protection against

the invading system, as shown in table 4. In all 70,0000 trials, E will not respond like A and B’s TLTPM. Table 5 displays the most cost-effective choices.

For various learning, weight range, and constant TLTPM size, table 6 analyzes the

synchronization periods of TLTPM and TPM-FPGA methods. When λ increases in all three rules, there is a propensity for the sync step to rise. In the λ  range of 5 to 15, Hebbian hardly requires any sync stages than that of the other two rules, but when the l rises, Hebbian uses more sync steps than the others.

The Anti-Hebbian harmonizing times of the TLTPM and CVTPM strategies are compared in table 7. The TLTPM’s sync interval is noticeably shorter than the CVTPM’s, as shown in table. The Anti-Hebbian, in comparison to others, requires less effort.

The necessary synchronizing duration for the 512-bit key generation with varying 786

synaptic ranges is shown in Figure 18. In large scale networks, the Random Walk rule

surpasses other learning techniques.

  1. In the Table 3, the specific methods represented by references [39], [40], [12], [41] and [42] should be presented.

Reply: As per the reviewer’s advice, in Table 3, the specific methods represented by these references are presented in the revised manuscript.

Liu et al. introduced recent conceptual conclusions on linked fractional-order recurrent neural networks' global synchronization. The synchronization delay is quite long, and thus is not an effective strategy for group synchronization.

Karakaya et al. presented a memristive chaotic circuit-based True random bit generator and its realization on an Embedded system. The genuine randomness of the produced random number is not tested in this article using the NIST test suite.

Dolecki and Kozera investigated the sync times achieved for network weights picked at random from either a homogeneous or a Gaussian distribution with varying standard deviations. The network's synchronization time is investigated as a function of various numbers of inputs and distinct weights pertaining to intervals of diverse widths. In possible to correlate networks with various weight intervals, the deviation of a Gaussian distribution is chosen based on this interval size, which is also a novel way for determining the distribution's variables.

Patidar et al. proposed a chaotic logistic map-based pseudo random bit generator. The genuine randomness of the produced random number is not examined in this work utilizing the 15 NIST test suite.

A chaotic PRBG system based on a non-stationary logistic map was developed by Liu et al. The researchers devise a dynamic approach to convert a non-random argument sequence into a random-like sequence. The changeable parameters cause the system's phase space to be disrupted, allowing it to successfully withstand phase space rebuilding attempts.

sized in the revised manuscript

Reviewer 2 Report

The paper describes artificial neural network based group key synchronization method for V2X applications.  It is hard to capture the main contribution of the paper.  The authors should provide clear explanation of their method and the presentation of the result in accademically appropriate format.

- Mutual Intelligent Transportation(MIT) should be defined first before use.

- The main contribution of the paper should be described clearly.  Lines 121--145 describes too many and wordy different contributions of the paper, so it is hard to find the main contribution.

- Some figures are too large compared to the information they provide.(Figures 3, 5, 11 etc.) 

- Reference in line 271is missing.

- There are many occurence of refering figures with "figure n" instead of "Figure n", sections with "section n" instead of "Section n".

- Acronyms should be explained when they are first used. In line 364 ANN is first used without refering what it means.

- The neural network model used in this paper should be clearly explained.

- In line 387, the meaning of flag g should be explained.

- No pesudo code for "Synchronization using B-Tree Framework" is provided whereas some pseudo code for "Synchronization using Ring Framework".

- Table 5 and 6 comes before Table 4.

- X, Y axis labels are missing in Figure 18.

- Conclusion should provide to-the-point summary of the work.

Author Response

The authors express their gratitude for the constructive comments and suggestions made by the esteemed reviewers. The followings are commentwise responses by the authors.

Reviewer #2:

The paper describes artificial neural network based group key synchronization method for V2X applications.  It is hard to capture the main contribution of the paper.  The authors should provide clear explanation of their method and the presentation of the result in accademically appropriate format.

Reply: As per the reviewer’s suggestion, the main contribution of the paper is incorporated in “Section 1: Introduction” the revised manuscript. The following paragraphs are incorporated in the revised manuscript.

The main contribution of this paper is to build a data fusion security architecture with variable levels of trust. This research also provides an efficient and effective information fusion security solution for various sources and multiple types of data sharing in V2X heterogeneous networks. For artificial neural synchronization-based rapid group key exchange, an area-based PKI infrastructure with speed enabled by a Graphics Processing Unit (GPU) is offered. To confirm that the suggested data fusion trust solution fulfill the rigorous delay requirements of V2X systems, a parametric test is performed. This article discusses the benefits of implementing an area-based key pre-distribution scheme. First, by lowering request volumes, the area-based key distribution strategy can minimize the mean delay. The mean delay can be reduced with a high accuracy prediction rate. Secondly, key distribution is a sort of information exchange procedure that includes the transmission of data as well as the essential security computations, whereas key requests/responses are a data type that is communicated in V2X heterogeneous networks. Third, GPU speeding reduces the time spent on security-related computation operations. Therefore, information exchange delay in V2X networks involving cars and RSUs is significantly lowered for important request/response.

The section provides an overview of the paper’s different significant contributions:

  1. This paper offers four levels of trust for the data amalgamation trust system which contain (i) There are no safety features, (ii) Verified source of data, (iii) The authenticity, as well as the integrity of the data source has been verified, (iv) The  data source has been confirmed, the data integrity has been checked, and the data content has been encoded.
  2. To achieve the third degree of belief for information fusion in V2X systems a key exchange system as well as safety computation activities has been implemented so  that the extra delay of implementing security methods must be taken into account,  as the delay criteria in this data amalgamation process is stringent because safety is a top priority.
  3. The delay has been decreased by minimizing key request updates with a position based key dispersal network.
  4. As per this research, the key exchange process must be developed by synchronizing a group of Triple Layer Tree Parity Machines (TLTPM). Rather than synchronizing individual TLTPM, the cluster members can utilize the identical key by cooperating among some of the selected skippers TLTPMs in logarithmic time.
  5. The secret key is created by exchanging very few parameters across an unprotected link while both sides use the neuronal coordination procedure.
  6. The proposed technique’s coordination duration for various learning rules are substantially fewer than the existing techniques.
  7. The key swap over strategies described by [8], [9], [10], [11], and [12] were investigated in the present study. This research focused on their weaknesses as well. To overcome the relevant problems, this article gives a TLTPM coordinating key agreement technique that results in a secret key with a flexible size.

The authors provide a clear explanation of their method in “Section 3: Proposed Methodology” and the presentation of the result in “Section 5: Results and Analysis” in academically appropriate format in the revised version of the manuscript.

- Mutual Intelligent Transportation(MIT) should be defined first before use.

Reply: As per the reviewer’s suggestion, Mutual Intelligent Transportation (MIT) is defined in “Section 1: introduction”. The following paragraph has been added in “Section 1: introduction”.

Mutual Intelligent Transport Systems (MITS) are transportation systems in which two or more ITS sub-systems (personal, car, roadside, and centralized) facilitate and deliver solution with higher quality and service level than if only one of sub-systems worked together. MITS will employ sophisticated ad hoc short-range communication technologies (such as ETSI ITS G5) as well as complementary wide-area communication technologies (such as 3G, 4G, and future 5G) to allow road vehicles to communicate with other vehicles, traffic signals, roadside infrastructural facilities, and other road users. Vehicle-to-vehicle (V2V), vehicle-to-infrastructure (V2I), and vehicle-to-person (V2P) interactions are all terms used to describe cooperative V2X systems.

- The main contribution of the paper should be described clearly.  Lines 121--145 describes too many and wordy different contributions of the paper, so it is hard to find the main contribution.

Reply: As per the reviewer’s suggestion, the main contribution of the paper is incorporated in “Section 1: Introduction” the revised manuscript. The following paragraphs are incorporated in the revised manuscript.

The main contribution of this paper is to build a data fusion security architecture with variable levels of trust. This research also provides an efficient and effective information fusion security solution for various sources and multiple types of data sharing in V2X heterogeneous networks. For artificial neural synchronization-based rapid group key exchange, an area-based PKI infrastructure with speed enabled by a Graphics Processing Unit (GPU) is offered. To confirm that the suggested data fusion trust solution fulfill the rigorous delay requirements of V2X systems, a parametric test is performed.

- Some figures are too large compared to the information they provide.(Figures 3, 5, 11 etc.) 

Reply: As per the reviewer’s advice, figures 3, 5, 11 are get resized.

- Reference in line 271is missing.

Reply: As per the reviewer’s suggestion, a reference is added in line 271.

- There are many occurence of refering figures with "figure n" instead of "Figure n", sections with "section n" instead of "Section n".

Reply: As per the reviewer’s advice, the occurrence of referring figures with "figure n" replaced with "Figure n", sections with "section n" replaced with "Section n"

- Acronyms should be explained when they are first used. In line 364 ANN is first used without refering what it means.

Reply: As per the reviewer’s suggestion, in line 364 the meaning of acronym ANN has been described in the revised manuscript.

- The neural network model used in this paper should be clearly explained.

Reply: As per the reviewer’s advice, the neural network model used in this paper is clearly explained using a synchronization algorithm in the revised version of the manuscript.

Sync. Mechanism:

Step 1: Assign the weight matrix Dq to any random vector value. Where, δqv,w {−λ,λ + 1, . . . , +λ}. When optimal synchronization has been achieved, repeat steps 2 through 5.

Step 2: The outcome of every concealed node is selected by a weight based on the current condition of the inputs. Equation 2 is being used to decide the outcome of the 1st concealed layer.

The result ξqv of the v-th concealed node is denoted by signum(κv) in equation 14. If κv = 0 is true, ξqv is mapped to 1, then binary outcome is produced. ξqv is assigned to +1. If κv > 0 is true, indicating that the concealed neuron is functioning. The concealed node is deactivated if the magnitude is ξqv= 1. Equation 3 shows how to do this.

Step 3: Calculate TLTPM’s ultimate result. TLTPM’s final outcome is computed by multiplying the hidden neurons in the last layer. The letter ζ stands for this (equation 4). The representation of the magnitude of ζis seen in equation 5. If TLTPM has one concealed neuron, ζ = ξq1. For 2μ1 distinct (ξq1 , ξq2,  ..., ξqμ) options, the Ζ value is the same.

Step 4: Once the results of two TLTPMs G and D coincide, ζG = ζD modifies the weights by using Hebbian learning algorithm, both TLTPMs will be learnt from each other (equation 6). 

If the weights for each TLTPM are the same, go to step 6.

Step 5: If the outcomes of the both TLTPMs differ, the weights cannot be altered, ζG ζD. Go to step 2 now.

Step 6: As cryptographic keys, use these coordinated weights.

- In line 387, the meaning of flag g should be explained.

Reply: As per the reviewer’s suggestion, the meaning of flag g is explained in the revised manuscript.

Here, flag g is used to indicate whether the output of of different user's are same or not. output of two are identical then flag set to 1. Otherwise, it is set to 0.

- No pesudo code for "Synchronization using B-Tree Framework" is provided whereas some pseudo code for "Synchronization using Ring Framework".

Reply: As per the reviewer’s advice, a synchronization algorithm is incorporated for “Synchronization using B-Tree Framework”. The following algorithm is added in the revised manuscript.

Sync. Mechanism:

Step 1: Assign the weight matrix Dq to any random vector value. Where, δqv,w {−λ,λ + 1, . . . , +λ}. When optimal synchronization has been achieved, repeat steps 2 through 5.

Step 2: The outcome of every concealed node is selected by a weight based on the current condition of the inputs. Equation 2 is being used to decide the outcome of the 1st concealed layer.

The result ξqv of the v-th concealed node is denoted by signum(κv) in equation 14. If κv = 0 is true, ξqv is mapped to 1, then binary outcome is produced. ξqv is assigned to +1. If κv > 0 is true, indicating that the concealed neuron is functioning. The concealed node is deactivated if the magnitude is ξqv= 1. Equation 3 shows how to do this.

Step 3: Calculate TLTPM’s ultimate result. TLTPM’s final outcome is computed by multiplying the hidden neurons in the last layer. The letter ζ stands for this (equation 4). The representation of the magnitude of ζis seen in equation 5. If TLTPM has one concealed neuron, ζ = ξq1. For 2μ1 distinct (ξq1 , ξq2,  ..., ξqμ) options, the Ζ value is the same.

Step 4: Once the results of two TLTPMs G and D coincide, ζG = ζD modifies the weights by using Hebbian learning algorithm, both TLTPMs will be learnt from each other (equation 6). 

If the weights for each TLTPM are the same, go to step 6.

Step 5: If the outcomes of the both TLTPMs differ, the weights cannot be altered, ζG ζD. Go to step 2 now.

Step 6: As cryptographic keys, use these coordinated weights.

- Table 5 and 6 comes before Table 4.

Reply: As per the reviewer’s suggestion, tables are rearranged properly in the revised manuscript.

- X, Y axis labels are missing in Figure 18.

Reply: As per the reviewer’s advice, X, Y axis labels are added in Figure 18.

- Conclusion should provide to-the-point summary of the work.

Reply: As per the reviewer’s suggestion, conclusion of this paper is rewritten in the revised version of the manuscript.

Conclusion:

This paper introduced a data amalgamation trust mechanism for V2X divergent net

works. To begin, a definition with four degrees of information amalgamation trust is

provided based on V2X application requirements. Then, in order to successfully execute

keys that serve as a foundation for all levels of information trust services in V2X networks, an area-based key sharing technique is created. Security-related calculations for data sharing, such as authenticated key exchange and extra data transmission, are taken into consideration and accelerated using GPGPU. It has been found that this trust mechanism significantly enhanced the quality of the current V2X network solution. The key shar ing procedure may be viewed as one of the information communication processes that require the maximum level of confidence. This paper suggested that to establish the security framework by distributing security keys, a key sharing network, such as a PKI, is necessary. First, the area-based key distribution technique can reduce mean latency by decreasing request quantities. With a high accuracy prediction rate, the mean delay may be decreased. Second, key distribution is a type of information exchange method that comprises data transmission as well as key requests/responses, whereas key requests/responses are a data type that is transmitted in V2X heterogeneous networks. Third, GPU acceleration cuts down on the time it takes to perform security-related computations. As a result, the time it takes for essential requests and responses to be exchanged in V2X networks including cars and RSUs is greatly reduced. However, unlike other approaches to traffic assumptions, the assumptions in this study is based on the perspective of a single vehicle rather than anticipating traffic flows for entire region. To put it another way, the prediction is based on the historical route information of a single vehicle. This paper also includes a discussion among the most significant key switch over methods. This analysis looks at their flaws along with providing TLTPM group syncing. A TLTPM-based key interchange method could result in quicker key interchange network. TLTPM can increase safety in real-world scenarios by modifying the vector size, as shown in this research. The effectiveness of the approach, along with current TPM-FPGA, CVTPM, and VVTPM procedures, is assessed. Furthermore, simulations are conducted to find the best τ1 τ2 τ3 N λ values for protecting against an intruding system. In varied learning rules, the TLTPM’s syncing duration are quicker than the existing TPM-FPGA, CVTPM, and VVTPM methods. The proposed technique’s shortcoming is that it does not use a deep learning algorithm to install vehicle route assumptions procedures in each vehicle end. It is instead deployed on the MITS central server. The average delay for key sharing, as indicated in Section 5, is largely dependent on the perfection of each vehicle’s path assumptions. However, for performance reasons, this method of assumptions should be installed on each vehicle end instead of the MITS’s central server. In the future, a deep learning technique can be used to deliver this type of assumption. Also, we intend to provide a more generic data communication trust model for information amalgamation in V2X that includes key sharing.

Reviewer 3 Report

Information Fusion in Autonomous Vehicle Using Artificial Neural Group Key Synchronization

 General Comments

This paper introduced a data amalgamation trust mechanism for V2X divergent networks.

The manuscript in my opinion is somewhat long and difficult to follow (28 pages). However, the methodology is well explained and the results are clear and well discussed. In my opinion, it could be publishable if some recommendations that I include below were followed.

Specific Comments

Line 36. DSRC (Dedicated short-range communications) should be defined the first time is cited.

Line 95. Group Key Agreement has been defined before, so you can you the abbreviation GKA.

Line 271, Reference “Teodoro et al” is wrong cited.

In my opinion, section 4 should be included in methodology section

Line 364. ANN (artificial neuronal networks) should be defined the first time is cited.

Line 381. It seems that “Synchronization using Ring Framework” could be section 3.1.

Line 408.   It seems that ·Synchronization using B-Tree Framework” could be section 3.2.

Line 454. Section 4 could be part of section 3 methodology.

Line 555, Should be renamed as Result and discussion. Discussion is mandatory in any manuscript, and results of the research are discussed in this section.

Note about References.

Authors should try to cite references from MDPI journal. Sensors, remote sensing, machines, applied sciences or sustainability are some of the long list of journals where topics similar to the manuscript are investigated.

Author Response

The authors express their gratitude for the constructive comments and suggestions made by the esteemed reviewers. The followings are commentwise responses by the authors.

Reviewer #3:

General Comments

This paper introduced a data amalgamation trust mechanism for V2X divergent networks.

The manuscript in my opinion is somewhat long and difficult to follow (28 pages). However, the methodology is well explained and the results are clear and well discussed. In my opinion, it could be publishable if some recommendations that I include below were followed.

Specific Comments

Line 36. DSRC (Dedicated short-range communications) should be defined the first time is cited.

Reply: As per the reviewer’s advice, DSRC (Dedicated short-range communications) is defined in ine 36.

Line 95. Group Key Agreement has been defined before, so you can you the abbreviation GKA.

Reply: As per the reviewer’s suggestion, Group Key Agreement has been abbreviated as GKA in line 95.

Line 271, Reference “Teodoro et al” is wrong cited.

Reply: As per the reviewer’s suggestion, correct reference is cited in line 271.

In my opinion, section 4 should be included in methodology section

Reply: As per the reviewer’s advice, Section 4 is discussed as a part of Section 3: Methodology section. Now Section 4 becomes Section 3.3.

Line 364. ANN (artificial neuronal networks) should be defined the first time is cited.

Reply: As per the reviewer’s advice, in line 364 the meaning of acronym ANN has been described in the revised manuscript.

Line 381. It seems that “Synchronization using Ring Framework” could be section 3.1.

Reply: As per the reviewer’s advice, “Synchronization using Ring Framework” is discussed under Section 3.1.

Line 408.   It seems that ·Synchronization using B-Tree Framework” could be section 3.2.

Reply: As per the reviewer’s suggestion, “Synchronization using B-Tree Framework” is discussed under Section 3.2.

Line 454. Section 4 could be part of section 3 methodology.

Reply: As per the reviewer’s advice, Section 4 is discussed as a part of Section 3. Now Section 4 becomes Section 3.3.

Line 555, Should be renamed as Result and discussion. Discussion is mandatory in any manuscript, and results of the research are discussed in this section.

Reply: As per the reviewer’s advice, the name of “Result and Analysis” section renamed as  “Result and Discussion” . Also, Section 5 becomes Section 4 in the revised manuscript.

Note about References.

Authors should try to cite references from MDPI journal. Sensors, remote sensing, machines, applied sciences or sustainability are some of the long list of journals where topics similar to the manuscript are investigated.

Reply: As per the reviewer’s advice, following references are incuded from the MDPI  Sensors journal.

Aliev, H.; Kim, H.; Choi, S. A Scalable and Secure Group Key Management Method for Secure V2V Communication. Sensors 2020, 20, 6137. https://doi.org/10.3390/s20216137

Alieve et al. presented a safe, lightweight, and scalable group key distribution and message encryption framework to tackle the secrecy of vehicle-to-vehicle (V2V) broadcasting. Leveraging scalable rekeying algorithms, the described group key management approach can handle diverse circumstances such as a node entering or leaving the group.

Han, B.; Peng, S.; Wu, C.; Wang, X.; Wang, B. LoRa-Based Physical Layer Key Generation for Secure V2V/V2I Communications. Sensors 2020, 20, 682. https://doi.org/10.3390/s20030682

Han et al. developed a LoRa-based physical key generation technique for protecting V2V/V2I interactions. The communication is based on the Long Range (LoRa) protocol, that may use the Received Signal Strength Indicator (RSSI) to produce secure keys over long distances.

Reviewer #4:

This study has developed a data fusion security infrastructure having varying levels of trust. The manuscript is well organised and focussed. However, I would suggest to authors to write a paragraph or two about the limitation of the experiment.

Reply: As per the reviewer’s advice, the limitation of the experiment is included in the conclusion section of the revised manuscript. The following paragraph for describing the limitation of the experiment is incorporated in the revised manuscript.

The proposed technique's shortcoming is that it does not use a deep learning algorithm to install vehicle route assumptions procedures in each vehicle end. It is instead deployed on the MITS central server. The average delay for key sharing, as indicated in Section 5, is largely dependent on the perfection of each vehicle's path assumptions. However, for performance reasons, this method of assumptions should be installed on each vehicle end instead of the MITS's central server. In the future, a deep learning technique can be used to deliver this type of assumption. Also, we intend to provide a more generic data communication trust model for information amalgamation in V2X that includes key sharing.

Authors should also consider merging some of the paragraphs as they tend to be to short to stand alone (e.g., line 185 to 187, line 514 to 515, etc.).

Reply: As per the reviewer’s suggestion, some of the paragraphs are merged as they tend to be to short to stand alone (e.g., line 185 to 187, line 514 to 515, etc.).

I think Figure 6 is unnecessarily too big and needs down sizing.

Reply: As per the reviewer’s advice, figure 6 is resized in the revised manuscript.

Reviewer 4 Report

This study has developed a data fusion security infrastructure having varying levels of trust. The manuscript is well organised and focussed. However, I would suggest to authors to write a paragraph or two about the limitation of the experiment. Authors should also consider merging some of the paragraphs as they tend to be to short to stand alone (e.g., line 185 to 187, line 514 to 515, etc.). I think Figure 6 is unnecessarily too big and needs down sizing.

Author Response

The authors express their gratitude for the constructive comments and suggestions made by the esteemed reviewers. The followings are comment wise responses by the authors.

Reviewer #4:

This study has developed a data fusion security infrastructure having varying levels of trust. The manuscript is well organised and focussed. However, I would suggest to authors to write a paragraph or two about the limitation of the experiment.

Reply: As per the reviewer’s advice, the limitation of the experiment is included in the conclusion section of the revised manuscript. The following paragraph for describing the limitation of the experiment is incorporated in the revised manuscript.

The proposed technique's shortcoming is that it does not use a deep learning algorithm to install vehicle route assumptions procedures in each vehicle end. It is instead deployed on the MITS central server. The average delay for key sharing, as indicated in Section 5, is largely dependent on the perfection of each vehicle's path assumptions. However, for performance reasons, this method of assumptions should be installed on each vehicle end instead of the MITS's central server. In the future, a deep learning technique can be used to deliver this type of assumption. Also, we intend to provide a more generic data communication trust model for information amalgamation in V2X that includes key sharing.

Authors should also consider merging some of the paragraphs as they tend to be to short to stand alone (e.g., line 185 to 187, line 514 to 515, etc.).

Reply: As per the reviewer’s suggestion, some of the paragraphs are merged as they tend to be to short to stand alone (e.g., line 185 to 187, line 514 to 515, etc.).

I think Figure 6 is unnecessarily too big and needs down sizing.

Reply: As per the reviewer’s advice, figure 6 is resized in the revised manuscript.

Round 2

Reviewer 1 Report

There are still some format problems which need to be revised, such as references, please double check them. 

Reviewer 2 Report

First round comments are all handled.

This manuscript is a resubmission of an earlier submission. The following is a list of the peer review reports and author responses from that submission.